# Inflammatory Response, Immunosuppression and Arginase Activity after Cardiac Surgery Using Cardiopulmonary Bypass

**DOI:** 10.3390/jcm11144187

**Published:** 2022-07-19

**Authors:** José María Rodríguez-López, José Luis Iglesias-González, Francisco Santiago Lozano-Sánchez, Miguel Ángel Palomero-Rodríguez, Pilar Sánchez-Conde

**Affiliations:** 1Department of Anesthesiology and Critical Care, Salamanca University Hospital, 37007 Salamanca, Spain; jiglesiashsr@hotmail.com (J.L.I.-G.); pconde@usal.es (P.S.-C.); 2Department of Angiology and Vascular Surgery, Salamanca University Hospital, 37007 Salamanca, Spain; lozano@usal.es; 3Department of Anesthesiology and Critical Care, La Paz University Hospital, 28046 Madrid, Spain; mapalomero@gmail.com

**Keywords:** arginase activity, CD3ζ chain expression, immunosuppression, cardiac surgery, cardiopulmonary bypass

## Abstract

Background: Major surgeries suppress patients’ cellular immunity for several days, but the mechanisms underlying this T-cell dysfunction are not well understood. A decreased L-Arginine (L-Arg) level may inhibit T-cell function. Arginase 1 (Arg 1) is induced after traumatic injury, leading to molecular changes in T cells, including decreased expression of cell surface T-cell receptors (TCRs) and a loss in CD3ζ chain expression. In this study, we examined the temporal patterns of CD3ζ expression and Arg 1 activity in patients undergoing cardiac surgery with cardiopulmonary bypass (CPB). Methods: We determined the CD3ζ chain expression; the Arg 1 activity; and the leukocyte, neutrophil and lymphocyte levels of patients on the day before surgery and at 24, 48 and 72 h after surgery. Results: Fifty adult patients scheduled for elective cardiac surgery with CPB were eligible for enrolment. Arginase activity was significantly increased between the day before surgery and at 24, 48 and 72 h after surgery (*p* < 0.01), and CD3ζ expression was significantly decreased between the day before surgery and at 24, 48 and 72 h after surgery (*p* < 0.001). We observed significant leukocytosis, neutrophilia and lymphopenia after surgery. Conclusions: The decreased CD3ζ chain expression could be due to the increased Arg 1 activity secondary to the activation of neutrophils in cardiac surgery under CPB. These findings could explain the limited immune-system-mediated organ damage resulting from systemic inflammatory response to major cardiac surgery with CPB.

## 1. Introduction

A systemic inflammatory response is triggered in patients undergoing cardiac surgery with cardiopulmonary bypass (CPB) as a result of the combination of surgical trauma, activation of blood components in the extracorporeal circuit, ischemia/reperfusion injury, endotoxin release and activation of immune cells. Immune responses offer protection to the organism from a variety of pathologic insults [1,2,3,4].

The immune system comprises two fundamental features that affect such protection by generating both innate, and adaptative or acquired responses to the insult. CPB impinges on these two elements of the immune response and induces quantitative and qualitative changes in the cellular and humoral constituents of the adaptive immune system, leading to temporary immunodeficiency [5].

Cardiac surgery under CPB is considered a type of major surgery, which are well known for suppressing cellular immunity for several days. T-cell dysfunction after surgery includes reduced cell surface T-cell receptors (TCRs) and diminished T lymphocyte proliferation responses, but the mechanisms underlying T-cell dysfunction have not been completely established [6,7,8,9,10].

L-Arginine (L-Arg) is a conditionally essential amino acid for adult mammals: that is, it must be supplied in the diet during certain physiological or pathological conditions, such as sepsis and trauma, in which the requirement exceeds the production capability. Myelomonocytic cells are equipped with extremely efficient mechanisms for destroying invading pathogens, and some subsets, known as myeloid suppressor cells (MSCs), are also highly efficient at suppressing activated T cells. One way in which these cells regulate T cells is by manipulating the metabolism of L-Arg, limiting L-Arg availability, which results in functional changes in T lymphocytes similar to that observed after trauma [11].

Arginase 1 (Arg 1), an enzyme expressed in different cell types, is induced after traumatic injury in MSCs and metabolizes L-Arg to ornithine, depleting L-Arg from the microenvironment and leading to distinct molecular changes in T cells, including decreased membrane expression of TCRs and a loss in CD3ζ chain expression, which play central roles in initiating the signal transduction cascade that leads to T-cell activation and proliferation.

This decrease in L-Arg availability after physical injury, coinciding with an induction of MSCs expressing Arg 1, may be linked to an increased susceptibility to complications after cardiac surgery under CBP. Although current research has focused on regulatory T lymphocytes as suppressors of autoimmune responses, powerful immunosuppression is also mediated by a subset of myeloid cells that enter the lymphoid organs and peripheral tissues during times of immune stress. If these MSCs receive signals from activated T lymphocytes, they block T-cell proliferation. MSCs use two enzymes involved in L-Arg metabolism to control T-cell responses: inducible nitric oxide synthase (iNOS), which oxidizes L-Arg in two steps that generate nitric oxide (NO) and citrulline, and Arg 1, which converts L-Arg into urea and L-ornithine. The induction of either enzyme alone results in a reversible block in T-cell proliferation. When both enzymes are induced together, peroxynitrites, generated by iNOS under conditions that limit L-Arg, cause activated T lymphocytes to undergo apoptosis [12,13]. Currently, the Arg 1 activity and CD3ζ chain expression in human mononuclear cells during cardiac surgery are unknown. In this regard, as a novelty, we examined the temporal patterns of CD3ζ chain expression and Arg 1 activity in a prospective, single-center observational study with 50 patients scheduled for cardiac surgery with CPB.

## 2. Materials and Methods

The protocol was approved by the Salamanca University Hospital Medical Ethics Committee. All patients signed written informed consent. Fifty adult patients scheduled for elective isolated coronary artery bypass grafting (CABG), valve surgery, or combined CABG and valve procedures with CPB were eligible for enrolment. The exclusion criteria were patients being treated with corticosteroids and/or immunosuppressants, patients with kidney or malignant diseases, patients with infections, patients undergoing urgent procedures, patients with left ventricle ejection fraction < 45% and patients with recent myocardial infarction (less than 10 days).

Similar anesthetic procedures and CPB management were carried out in all patients. Premedication consisted of standard cardiac drugs and oral alprazolam (0.5 mg) administered 30 min prior to induction. Anesthesia was induced with i.v. administration of 0.03–0.04 mg·kg^−1^ of midazolam, 20–30 µg·kg^−1^ of fentanyl and 0.2–0.3 mg·kg^−1^ of etomidate. Muscular relaxation was achieved with 0.6 mg·kg^−1^ of rocuronium, and mechanical ventilation was provided at 50% oxygen/air. Anesthesia was maintained with i.v. administration of propofol (3–4 mg·kg^−1^·h^−1^), remifentanil (0.15–0.2 µg·kg^−1^·min^−1^) and cisatracurium (0.2 mg·kg^−1^·h^−1^). Before the induction of anesthesia, five-lead electrocardiogram and pulse oximetry were routinely used for monitoring, and an arterial line was secured (femoral artery) for continuous invasive blood-pressure monitoring. After the induction of anesthesia, an internal jugular venous catheter was inserted for central venous-pressure measurement. Extra corporeal circulation (ECC) was performed under moderate hypothermia (temperature between 30 °C and 32 °C) at a continuous flow of 60–80 mL·kg^−1^·min^−1^ using a SORIN CP 5 CPB machine fitted with membrane oxygenators. The priming solution for the circuit consisted of 500 mL of Ringer’s lactate and 500 mL of gelatin. Cold cardioplegia, consisting of cold blood and crystalloid in the ratio of 4:1, was used for all patients. Prior to vascular cannulation, 300 IU·kg^−1^ of heparin was administered i.v. and the activated clotting time (ACT) was determined after 3 min to achieve an ACT > 480 s. At the end of ECC, protamine sulphate in a ratio of 1.5:1 to heparin was administered. After surgery, the patients were transferred to the intensive care unit (ICU) and weaned from ventilation when they became hemodynamically stable and were re-warmed. Upon transfer to the ICU, infusions of remifentanil continued at 0.05–0.1 µg·kg^−1^·min^−1^ for sedation until the patient recovered spontaneous ventilation.

Blood samples were collected at different time points (T) in order to assay data corresponding to the arginase activity; the CD3ζ expression; and the leukocyte, neutrophil and lymphocyte levels of the patients: the day before surgery (T0), 24 h after surgery (T1), 48 h after surgery (T2) and 72 h after surgery (T3).

Mononuclear cells were isolated from fresh human peripheral blood by dextran sedimentation and centrifugation on Ficoll–Paque density gradients [14]. The mononuclear cells were saved and then washed twice with phosphate-buffered saline (PBS). The monocytes were depleted by culture dish adherence. After overnight incubation at 37 °C, the nonadherent cells (lymphocytes) were washed with PBS and collected. The lymphocyte samples were then kept at −20 °C until Western blot analysis.

The quantification of CD3ζ was determined by Western blot analysis following the electrophoresis of equally loaded samples of isolated peripheral blood lymphocytes in each gel electrophoresis lane, as previously described [13].

The measurement of arginase activity was performed according to the protocol previously described in the literature [15]. The sonication protocol for cell lysis was followed. We measured the hydrolysis of arginine by converting L-Arginine to urea. One unit of enzymatic activity was defined as the amount of enzyme needed to catalyze the formation of 1 µmol of urea·min^−1^.

The normally distributed data (tested using the Kolmogorov–Smirnov test and Shapiro–Wilk test) were analyzed using the Student two-tailed *t*-test for independent samples to compare the leukocyte, neutrophil and lymphocyte counts; the arginase activity; and the CD3ζ expression from the day before surgery with those at 24, 48 and 72 h after surgery. Pearson correlation coefficient was used to assess the linear correlation between arginase activity and CD3ζ expression 24 h after surgery. The sample size needed to detect a coefficient of determination (r) ≥ 0.5, based on earlier observations^9^, to obtain a β level greater than 85% and an α level less than 5% was estimated to be fifty patients. All statistical analyses were performed using IBM SPPS Statistics for Windows Version 20.0 (Armonk, NY, USA: IBM Corp.). A *p* value < 0.05 was considered statistically significant.

## 3. Results

Fifty patients were enrolled in this prospective observational study. The mean age was 70.4 ± 1.6 years, and 66% (n = 33) were male. The majority of patients underwent valve procedures (n = 36). A more detailed description of the patient characteristics are summarized in Table 1.

There were significant increases in arginase activity between the day before surgery and at 24, 48 and 72 h following surgery (Figure 1A), with a maximum increase in arginase activity at 24 h after surgery (3.24 ± 0.36 mU·mg^−1^·min^−1^; *p* < 0.01) (Table 2).

We found a significant decrease in CD3ζ expression between the day before surgery and at 24, 48 and 72 h after surgery (Figure 1B), with a minimum CD3ζ expression at 72 h after surgery (9.8 ± 13.48%; *p* < 0.001) (Table 2).

There were increases in white blood cell counts after surgery (Figure 2A), with significantly higher leukocytosis at 24 and 48 h after surgery (11.67 ± 5.56 × 10^3^·µL^−1^ and 12.90 ± 6.38 × 10^3^·µL^−1^, respectively; *p* < 0.05) (Table 2).

The neutrophil counts significantly increased after surgery, with higher neutrophilia at 48 h after surgery (10.06 ± 0.76 × 10^3^·µL^−1^; *p* < 0.001) (Table 2).

The lymphocyte counts decreased after surgery (Figure 2B), with significant lymphopenia at 24 and 48 h after surgery (0.83 ± 0.07 × 10^3^·µL^−1^ and 1.13 ± 0.07 × 10^3^·µL^−1^, respectively; *p* < 0.01) (Table 2).

We found a negative linear correlation between the levels of arginase activity and CD3ζ expression in the first 24 h after surgery (r = −0.79; *p* = 0.042) (Figure 3).

## 4. Discussion

Cardiac surgery with CPB not only induces a proinflammatory response but also induces an immunosuppression state. The cellular and humoral constituents of the adaptive immune system undergo changes in both function and number after CPB. The concern is that postoperative morbidity and mortality related to infection are partially due to losses in T and B cells, and immunoglobulin consumption resulting from CPB. Our study examines the effect of CPB on adaptative immunity and, specifically, T-cell function. T cells are lymphocytes that develop in the thymus; then, develop receptors for T-cell antigens; and finally, differentiate into two major peripheral T-cell subsets, one of which expresses the CD4+ marker (helper cells) and the other of which expresses the CD8+ marker (cytotoxic cells). T helper cells (T_H_) play a central role in the initiation and regulation of the acquired immune response and provide “help” in the form of cytokine secretion: T_H_1 cytokines (interferon-_ϒ_ and tumor necrosis factor-α) and T_H_2 cytokines (interleukin-4 (IL-4), interleukin-13 (IL-13), interleukin-10 (IL-10) and transforming growth factor-β (TGF-β)) [5,16]. Intense burdens on the immune system induce profound immunosuppression of adaptive responses. The principal mediator of this suppression is MSCs. These cells are generated from bone-marrow hemopoietic precursors in response to several cytokines and act as sensors of T-cell activation during heightened immune responses [17].

MSCs exploit the metabolism of L-Arg to render lymphocytes unresponsive to antigen stimulation. L-Arg is metabolized by two enzymes: iNOS and Arg1. Arg-1 is induced by T_H_2 cytokines [18,19]. In MSCs, the coordinated induction of Arg 1 and a cationic amino acid transporter by IL-4 and IL-13 conveys L-Arg from the extracellular milieu into the cells, where it is degraded. The depletion of L-Arg in the microenvironment would block TCR signaling and modulate T-cell function at an early stage of lymphocyte activation. CD3ζ is the main signal-transduction component of the TCR and is required for correct assembly of the receptor complex. Losses in CD3ζ expression are the only Arg1-triggered mechanism described so far that has been proven to have direct relevance to T-cell function and proliferation. Indeed losses in CD3ζ chain expression in peripheral blood lymphocytes have been reported in patients under conditions that might lead to Arg1 activation, including tumor growth, overwhelming infections and trauma [20,21,22,23].

In our study, we found a significant increase in Arg 1 activity associated with a significant decrease in CD3ζ expression from 24 to 72 h after surgery and show the correlation between Arg 1 activity and CD3ζ expression 24 h after surgery. Our results may suggest that the increase in Arg 1 activity, likely from an increase in neutrophil count, could be used by MSCs to inhibit T-cell responses to antigen. In fact, these MSCs incorporated and readily consumed L-Arg from the extracellular environment, and they inhibited CD3ζ expression after its TCR-signaling-induced internalization by antigen-stimulated T cells, thereby impairing the function and proliferation of T cells. This mechanism seems to be involved in the physiological anti-inflammatory response that limits immune-system-mediated organ failure resulting from the systemic inflammatory response secondary to cardiac surgery with CPB that makes patients susceptible to perioperative infections, with septic shock being a possible, frequently fatal result. To our knowledge, this is the first study that correlate arginase activity with CD3ζ expression following cardiac surgery. Colagrande et al. [24] determined that the use of L-Arg added to a cardioplegia solution in coronary artery bypass graft surgery patients with CPB is safe, is able to reduce myocardial stress and has a protective effect.

We observed an increase in leukocyte and neutrophil counts but not in lymphopenia after CPB. Most investigators consider neutrophils to play the central role in the tissue and organ injuries that result from CPB [25]. Their lymphocyte concentrations decrease over the duration of CPB, and this decrease remains until the third day after surgery. DePalma et al. [26] reported that a decrease in lymphocyte counts remains for around 3–7 days after cardiac surgery with CPB. This reduction in lymphocyte number combined with an impairment in white-cell phagocytosis and an inhibition of CD3ζ expression results in a weakened cellular immune response and an increased susceptibility of post-CPB patients to infections.

One limitation of our study is not having considered the metabolism of L-Arg by iNOS. The induction of iNOS is controlled mainly by T_H_1 cytokines, and its enzymatic activity is sustained over a long period of time. The induction of iNOS alone in MSCs, with subsequent release of NO, is responsible for the inhibition of T-cell responses and should be considered as a mechanism of immunomodulation [27].

## 5. Conclusions

Our research suggests that the decreased CD3ζ chain expression found in our patients could be due to the increased Arg 1 activity 24 h after cardiac surgery with CPB and with the depletion of L-Arg. This increased Arg 1 activity after surgery is due to the activation of neutrophils and the release of cytokines after CPB. These findings could explain the immunosuppression state that occurs after cardiac surgery with CPB, but it is tempting to suggest that systemic inflammatory response after cardiac surgery with CPB could be accompanied by an anti-inflammatory process in which Arg 1 activity could play an important role, preventing excessive inflammatory response, as a possible mediator of septic complications after cardiac surgery. Perhaps a question in the clinical setting is whether the administration of L-Arg during the perioperative nutrition period could ameliorate the immunosuppression state and perhaps improve the outcomes after cardiac surgery with CPB. Further studies should be performed to establish causality between the above data in an attempt to provide a meaningful and statistically valid connection, and a causal relationship between increased Arg 1 activity and decreased CD3ζ chain expression to explain the temporary immunosuppression state that occurs after a major surgery.

## Figures and Tables

**Figure 1 jcm-11-04187-f001:**
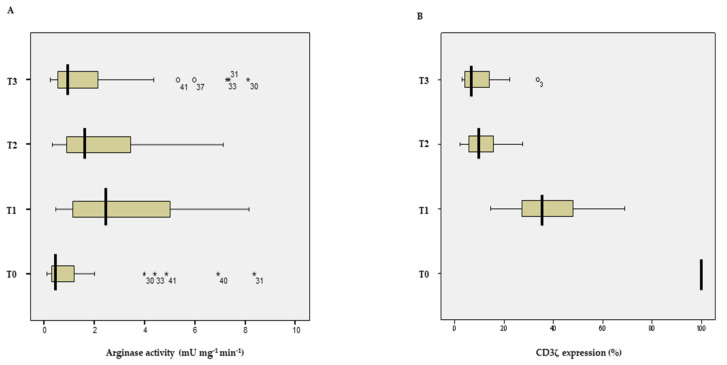
(**A**) Arginase activity (mU·mg^−1^·min^−1^) and (**B**) CD3ζ expression (%) at different time points. Data are expressed as median value of the distribution (thick vertical line) and interquartile values. Data away from a distance greater than 1.5 times the interquartile range are represented by a circle and data away from a distance greater than 3 times the interquartile range are represented by an asterisk. Time points: T0, the day before surgery; T1, 24 h after surgery; T2, 48 h after surgery; T3, 72 h after surgery.

**Figure 2 jcm-11-04187-f002:**
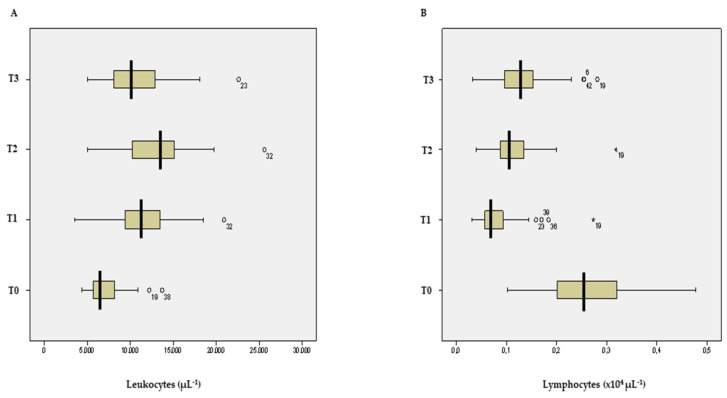
(**A**) Leukocytes (µL) and (**B**) Lymphocytes (×10^3^ µL^−1^) at different time points. Data are expressed as median value of the distribution (thick vertical line) and interquartile values. Data away from a distance greater than 1.5 times the interquartile range are represented by a circle and data away from a distance greater than 3 times the interquartile range are represented by an asterisk. Time points: T0, the day before surgery; T1, 24 h after surgery; T2, 48 h after surgery; T3, 72 h after surgery.

**Figure 3 jcm-11-04187-f003:**
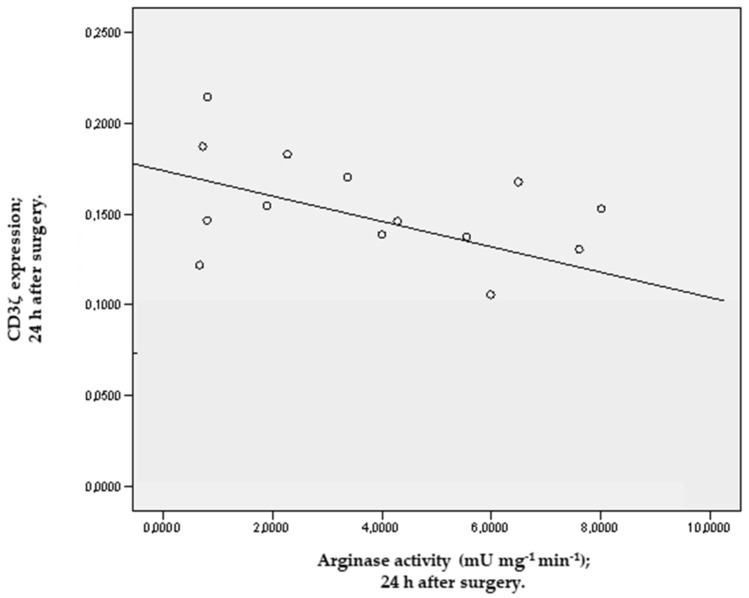
Linear correlation between Arginase activity/CD3ζ; 24 h after surgery.

**Table 1 jcm-11-04187-t001:** Patients characteristics (n = 50).

Age (years)	70.4 ± 1.6
Sex (M/F)	33/17
Hypertension	35 (70%)
Diabetes Mellitus	12 (24%)
Dyslipidemia	39 (78%)
Myocardial Ischemia	14 (28%)
Heart Valve Disease	39 (78%)
Atrial Fibrillation	17 (34%)
COPD	22 (44%)
Smoker	22 (44%)
Aspirin	24 (48%)
Oral anticoagulants	17 (34%)
ACE inhibitor	27 (54%)
Statin	34 (68%)
Oral antidiabetic agents	12 (24%)
Βeta blockers	27 (54%)
Loop diuretics	22 (44%)
Valve surgery	36 (72%)
CABG	11 (22%)
Valve surgery + CABG	3 (6%)
CPB (min)	116.9 ± 5.4
Aortic clamping (min)	88.5 ± 4.5

Data are presented as means ± standard deviations, absolute numbers and percentages. Abbreviations: M, men; F, female; COPD, chronic obstructive pulmonary disease; ACE, angiotensin converting enzyme; CABG, coronary artery bypass grafting; CPB, cardiopulmonary bypass.

**Table 2 jcm-11-04187-t002:** Time course of distinct parameters measured in the blood from patients undergoing cardiac surgery with cardiopulmonary bypass.

Time Course	T0	T1	T2	T3
Arginase activity (mU·mg^−1^·min^−1^)	1.20 ± 0.27	3.24 ± 0.36 **	2.29 ± 0.28 ***	1.83 ± 0.32 ***
CD3ζ expression (%)	100	7.58 ± 22.52 ****	11.10 ± 12.38 ****	9.8 ± 3.48 ***
Leukocytes (×10^3^ µL^−1^)	7.09 ± 3.18	11.67 ± 5.56 *	12.90 ± 6.38 *	10.63 ± 5.59
Neutrophils (×10^3^ µL^−1^)	4.46 ± 0.68	9.6 ± 0.53 **	10.06 ± 0.76 ***	8.17 ± 0.73 **
Lymphocytes (×10^3^ µL^−1^)	2.71 ± 0.14	0.83 ± 0.07 **	1.13 ± 0.07 **	1.32 ± 0.08

Data are presented as means ± standard deviations and percentages. Abbreviations: T0, the day before surgery; T1, 24 h after surgery; T2, 48 h after surgery; T3, 72 h after surgery. * *p* < 0.05; ** *p* < 0.01; *** *p* < 0.001; **** *p* < 0.0001.

## Data Availability

The data presented in this study are available from the corresponding author upon request, though they are restricted to investigators based in academic institutions.

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
