# Peer review of "Inflammatory Response, Immunosuppression and Arginase Activity after Cardiac Surgery Using Cardiopulmonary Bypass"

_jcm, 2022, doi:10.3390/jcm11144187_

Round 1

Reviewer 1 Report

With this study, the authors want to investigate one of the mechanisms that determine the inflammatory response, the suppression of the immune system following cardiac surgery. In particular, the authors identify among these mechanisms the Arginine deficiency as a possible cause of the reduced activity of T cells. This study on 50 patients measures the activity of Arginase, CD3 expressions, and the number of Neutrophils and lymphocytes. The results are clearly reported, and the conclusions are relevant to the results. However, some considerations are necessary to improve the paper:

- In Table 1, the mean time of CPB and aortic clamping could also be reported with the patient data.

- The authors could analyze whether the measured parameters correlate with the patient's CPB time and the aortic clamping times that determine the response to ischemic and reperfusion injury.

- L-Arginine is also metabolized by inducible nitric oxide synthase; iNOS is activated by the inflammatory response and in the heart due to ischemic injury and reperfusion. The authors should also consider a possible mechanism of NO immunomodulation produced by iNOS as a cause of the reduction of Arginine and a possible cause of the reduction of T-lymphocyte activity. The authors should better develop this aspect of NO involving immunomodulation.

Author Response

Dear Reviewer 1.  I send you my reply.

Thanks again for your comments.

Prof. Dr. Jose Maria Rodriguez-lopez

Reviewer 2 Report

In this study, authors investigated selected responses after cardiac surgery. Study is interesting, however, it has several concerns that should be addressed:

In Introduction section, more emphasis with detailed explanation should be put into exact literature gap this study is filling and accompanying novelty of the study, as that is not clear enough in current manuscript.

Rationale as why patients with ventricle ejection fraction < 45% were excluded from the study

Tables should be adapted to fit into journal guidelines (with font etc.)

In Table 1, at least percentages could be added

Figures are presented with low resolution and they are extremely not reader-friendly. Moreover, they are somewhat confusing and hard to understand. I suggest replacement with more appropriate figures. Finally, figure captions are incorporated into figures itself.

Figure 4 is not needed in the Discussion.

Limitations of this study should be emphasized better in the Discussion section, as well as more clinical implications that can be derived from the results.

Author Response

Dear Reviewer 2. I send you the reply.

Thanks again for your comments.

Prof. Dr. Jose Maria Rodriguez

Round 2

Reviewer 2 Report

No further comments.